# Simulating Extracellular Glucose Signals Enhances Xylose Metabolism in Recombinant *Saccharomyces cerevisiae*

**DOI:** 10.3390/microorganisms8010100

**Published:** 2020-01-10

**Authors:** Meiling Wu, Hongxing Li, Shan Wei, Hongyu Wu, Xianwei Wu, Xiaoming Bao, Jin Hou, Weifeng Liu, Yu Shen

**Affiliations:** 1State Key Laboratory of Microbial Technology, Institute of Microbial Technology, Shandong University, Qingdao 266237, China; 2State Key Laboratory of Biobased Material and Green Papermaking, School of Bioengineering, Qi Lu University of Technology, Jinan 250353, China

**Keywords:** budding yeast, xylose metabolism, glucose signaling pathway, PKA, Rgt1

## Abstract

Efficient utilization of both glucose and xylose from lignocellulosic biomass would be economically beneficial for biofuel production. Recombinant *Saccharomyces cerevisiae* strains with essential genes and metabolic networks for xylose metabolism can ferment xylose; however, the efficiency of xylose fermentation is much lower than that of glucose, the preferred carbon source of yeast. Implications from our previous work suggest that activation of the glucose sensing system may benefit xylose metabolism. Here, we show that deleting cAMP phosphodiesterase genes *PDE1* and *PDE2* increased PKA activity of strains, and consequently, increased xylose utilization. Compared to the wild type strain, the specific xylose consumption rate (r_xylose_) of the *pde1Δ pde2Δ* mutant strains increased by 50%; the specific ethanol-producing rate (r_ethanol_) of the strain increased by 70%. We also show that *HXT1* and *HXT2* transcription levels slightly increased when xylose was present. We also show that *HXT1* and *HXT2* transcription levels slightly increased when xylose was present. Deletion of either *RGT2* or *SNF3* reduced expression of *HXT1* in strains cultured in 1 g L^−1^ xylose, which suggests that xylose can bind both Snf3 and Rgt2 and slightly alter their conformations. Deletion of *SNF3* significantly weakened the expression of *HXT2* in the yeast cultured in 40 g L^−1^ xylose, while deletion of *RGT2* did not weaken expression of *HXT2*, suggesting that *S. cerevisiae* mainly depends on Snf3 to sense a high concentration of xylose (40 g L^−1^). Finally, we show that deletion of Rgt1, increased r_xylose_ by 24% from that of the control. Our findings indicate how *S. cerevisiae* may respond to xylose and this study provides novel targets for further engineering of xylose-fermenting strains.

## 1. Introduction

Lignocellulosic biomass is considered as an abundant, renewable, and environment-friendly material for biofuels and chemicals production. Converting the two most abundant sugars in hydrolysates of lignocellulosic materials, glucose and xylose, can increase both the environmental and economic benefits of lignocellulose as a source for biofuels [1,2]. *Saccharomyces cerevisiae* has been considered a highly competitive cell factory for conversion of lignocellulosic materials to biofuels and chemicals because it is a generally recognized as safe (GRAS) microorganism by the U.S. Food and Drug Administration, has strong glucose-metabolizing capacity, and has been well-studied. However, *S. cerevisiae* cannot utilize xylose because of its inability to process xylose in its metabolic pathways [1,3].

In recent decades, continuous efforts have been made to construct xylose-utilizing *S. cerevisiae* and improve the xylose metabolic capacity of these recombinant strains. Xylose reductase (XR) and xylitol dehydrogenase (XDH) of *Scheffersomyces stipitis* or xylose isomerases (XI) of bacteria and fungi have been introduced into *S. cerevisiae* to build pathways for xylose metabolism. In recombinant *S. cerevisiae* strains, xylose is transported by hexose transporters and metabolized in sequence through the XR-XDH or XI, pentose phosphate, and glycolysis pathways to produce pyruvate, which is then converted to ethanol and other products [3,4,5]. Therefore, the genes of xylulokinase and the non-oxidative part of the pentose phosphate pathway (PPP) were overexpressed to enhance the downstream flux of xylose metabolism [6,7,8].

Unfortunately, the attempts of metabolic engineering mentioned above are far from enough, and the adaptive evolution in the medium with xylose as the sole carbon source is necessary to produce a *S. cerevisiae* strain that can efficiently utilize xylose [8,9,10,11,12]. Many researchers have endeavored to reveal the differences in the “omics” between the evolved strains (with high xylose utilization capacity) and their parents (with low xylose utilization capacity) [11,13,14,15], as well as the differences in the “omics” between the strains cultured in xylose and in glucose [16,17]. The results suggested that an important reason that limited the xylose fermentation rate is the fact that *S. cerevisiae* lacks a signaling pathway to recognize xylose as a carbon source and regulate the cells to convert to a state that promotes xylose utilization. For another, our previous work suggests that extracellular glucose signals can promote xylose utilization. In a strain that could transport xylose but not glucose intracellularly, we observed that xylose metabolism was enhanced by the presence of extracellular glucose [18] (Figure A1). Extensive studies on glucose signaling pathways and their controls on glucose metabolism showed that efficient hexose transporters and glycolysis, which are the factors for efficient xylose metabolism, depends on activation of glucose signaling pathways. Unfortunately, how these signaling pathways may respond to xylose is not clear.

There are two signaling pathways that respond to extracellular glucose [19]. The first is the cAMP-PKA pathway (Figure 1A) where the transmembrane protein, Gpr1, undergoes an allosteric effect when extracellular glucose or sucrose binds to it. Then, the allosteric Gpr1 stimulates the transition of the small G protein Gpa2 from an inactive state (binding with GDP) to an active state (binding with GTP). The GTP-bound Gpa2 activates adenylate cyclase Cyr1 which catalyzes the conversion of ATP to cAMP and subsequently increases the intracellular levels of cAMP. cAMP binds to the regulatory subunit Bcy1 of PKA and exposes the active site of Tpk1/Tpk2/Tpk3, thus activating PKA. Meanwhile, the cAMP-PKA pathway is also regulated by an intracellular protein, Ras. Intracellular glucose and its metabolites stimulate the GDP-bound Ras (inactive) to convert to GTP-bound Ras (active). The active Ras can also increase Cyr1 activity and consequently, PKA activity. Additional proteins that affect the pathway are two cAMP phosphodiesterases, Pde1 and Pde2, that catalyze cAMP to AMP to prevent the overactivation of PKA [19]. The active PKA up-regulates glycolysis and down-regulates gluconeogenesis at both transcriptional and translational levels [20,21]. Increased activity of PKA benefits xylose utilization because xylose metabolism occurs through the PPP and glycolysis pathway.

In the Rgt2/Snf3-Rgt1 pathway, which also responds to extracellular glucose (Figure 1B), high and low levels of extracellular glucose are sensed by the membrane-spanning proteins Rgt2 and Snf3, respectively [22]. The signal is then relayed into the cell, leading to the phosphorylation-dependent degradation of Mth1 and Std1, which are co-repressors necessary for the DNA-binding of Rgt1. The elimination of Mth1 and Std1 exposes Rgt1 to phosphorylation, which releases it from gene promoters and derepresses the expression of target genes, including the ones encoding hexose transporters [19,23]. Lacking a xylose transporter, the transport of xylose in *S. cerevisiae* depends on these hexose transporters (HXT) [24]. Thus, the repression of Rgt1 is expected to negatively affect hexose transporter genes and thus xylose utilization. In a recent work, researchers were able to show that extracellular xylose can induce the fluorescent signal of the green fluorescent protein gene (*yEGFP3*) expressed under the control of the *HXT1*, *2*, and *4* promoters. Their data suggest that the Rgt2/Snf3-Rgt1 pathway responds to extracellular xylose, although the mechanistic details are still not clear [25].

The aims of the present work were to determine how signals from extracellular glucose can enhance xylose utilization and the mechanism of how *S. cerevisiae* can sense extracellular xylose. Here, we demonstrated that the PKA level in cells cultured on xylose was much lower than that of cells cultured on glucose. Additionally, expression of the *Gpa2^G132V^* allele or knock outs of both *PDE1* and *PDE2* increased the PKA activity, regulated expression of carbohydrate metabolism genes, and increased the xylose consumption rate of the *S. cerevisiae* strain. We also explored how *S. cerevisiae* may sense extracellular xylose by examining the transcriptional levels of *HXT1* and *HXT2* in *rgt2Δ* and *snf3Δ* strains cultured in media with different levels of glucose and xylose. The results demonstrate that Rgt2 and Snf3 can only weakly respond to high concentrations of xylose and slightly increase the expression of transporter genes. Then, we demonstrated that deleting Rgt1 improved the expression of transporter genes and xylose utilization. Our work will improve our understanding of the effect of xylose on the major glucose signaling pathways in *S. cerevisiae* and demonstrates promising strategies for delivering simulated glucose signals to induce *S. cerevisiae* cells to modify their gene-expression state to a state fit for xylose utilization.

## 2. Materials and Methods

### 2.1. Strains and Plasmids

All *S. cerevisiae* strains and plasmids are listed in Table 1. The primers are listed in Table A1.

Strain BSL01 (BSPC039 [12] derivative; pJX7 [26]) was used as the parent strain and control. The strain BSL06(*GPA2^G132V^*) was obtained by integrating *GPA2^G132V^* at the *GRE3* site of BSL01. The fragment *GRE3-ty1-loxP-KanMX-loxP-TEF1p-GPA2^G132V^-PGKt-GRE3-ty2* was used for the integration. The homologous arms *GRE3-ty1* and *GRE3-ty2* were cloned from the CEN.PK113-5D [27] genome. The selection marker *loxP-KanMX-loxP* was cloned from plasmid pUG6 [28]. The *GPA2^G132V^* ORF fragment was obtained by fusion PCR, using the CEN.PK113-5D genome as a template, and then treated with *Bam*HI and *Sbf*I and ligated between *TEF1p* and *PGKt* of plasmid pJFE1 [26]. Then, the expression cassettes *TEF1p-GPA2^G132V^-PGKt*, *GRE3-ty1*, *GRE3-ty2*, and *loxP-KanMX-loxP* were ligated by fusion PCR to obtain the full fragment *GRE3-ty1-loxP-KanMX-loxP-TEF1p-GPA2^G132V^-PGKt-GRE3-ty2*. The *KanMX* marker was discarded by transferring plasmid YEp-CH [26] into the strains and inducing the expression of the Cre recombinase. Thus, the *KanMX4* cassette between the two loxP sequences was removed by the recombination between the two loxP sites. The construction process of BSL08(*RAS2^G19V^*) was similar to that of BSL06(Gpa2^G132V^).

The deletions of genes *PDE1*, *PDE2*, *RGT1*, *SNF3*, and *RGT2* were all performed by replacing the target gene with the *KanMX4* cassette via homologous recombination, and then the *KanMX4* was removed as mentioned above. The integration arms of genes were all cloned from the CEN.PK113-5D genome.

### 2.2. Cultivation Conditions and Batch Fermentation

BSL01 and its derivatives were cultured in SD-URA medium, which contained 1.7 g L^−1^ yeast nitrogen base (YNB, Sangon, Shanghai, China), 5 g L^−1^ ammonium sulfate (Sangon, Shanghai, China), 0.77 g L^−1^ CSM-URA (MP Biomedicals, Solon, OH, USA), and 20 g L^−1^ glucose or xylose as the carbon source. BSWW1 (*snf3Δ*) and BSWW2 (*rgt2Δ*) were cultured in YPD medium. For selection of transformants, 800 mg L^−1^ G418 or 400 mg L^−1^ hygromycin B was added into the medium when necessary. The fermentations were performed in SD-URA medium with 20 g L^−1^ xylose and/or 20 g L^−1^ glucose at 30 °C. A single colony from a plate of SD-URA medium was incubated in the liquid medium and collected in the middle of exponential growth to inoculate 100-mL flasks containing 40 mL fermentation medium. The initial biomass was 0.23 g DCW L^−1^ and pH was 4.5. The fermentation was performed at 200 rpm, 30 °C.

### 2.3. Determination of cAMP

Preparation of samples was performed as instructed in the manual of the cAMP Direct Immunoassay Kit (BioVision, Milpitas, CA, USA). Cells in the middle of exponential growth were collected by centrifugation at 6000 rpm for 5 min at 4 °C. The 80 mg (DCW) cells were cooled in liquid nitrogen and resuspended with 500 μL HCl (0.2 M pre-cooled to 4 °C) and 0.6 g acid-washed glass beads (φ = 0.5 mm). Then cells were lysed using a FastPrep cell homogenizer (Thermo Savant, Carlsbad, CA, USA) at a speed of 5000 rpm for 30 s and repeated twice. The homogenate was centrifuged at 13,000 rpm for 5 min at 4 °C, and the cAMP levels in this supernatant were determined by the cAMP Direct Immunoassay Kit (BioVision, Milpitas, CA, USA). The t-tests were applied to evaluate the differences between means.

### 2.4. Assay of Trehalase Activity

The preparation of crude enzyme samples and the assay of trehalase activity were performed based on a previously reported method [29]. Cells were collected by centrifugation at 6000 rpm for 5 min at 4 °C. The 40 mg (DCW) cells were cooled in liquid nitrogen and resuspended with 500 μL HEPES (pH 7.0) and 0.8 g acid-washed glass beads (φ = 0.5 mm). These cells were lysed using a FastPrep cell homogenizer (Thermo Savant, Carlsbad, CA, USA) at a speed of 5000 rpm for 30 s and repeated five times. The homogenate was centrifuged at 13,000 rpm for 5 min at 4 °C, and the supernatant was used as the crude enzyme sample. The total cellular protein concentration was measured using a BCA protein assay reagent kit (Sangon Biotech Co., Ltd., Shanghai, China). Trehalase activities were determined at 30 °C by measuring the glucose produced by hydrolysis of trehalose. Reactions contained 10 μL extract and 140 μL water in 100 μL substrate buffer (250 mM trehalose in 50 mM HEPES buffer, pH 7, with 125 μM CaC1_2_). They were incubated at 30 °C for 30 min and then immediately halted by boiling for 3 min at 100 °C. The amount of glucose produced was determined by a d-Glucose (GOPOD Format) Assay Kit (Megazyme, Dublin, Ireland). Specific activity of trehalase was expressed as nmol glucose liberated min^−1^ (mg protein)^−1^. The t-tests were applied to evaluate the differences between means.

### 2.5. Analysis of Metabolites

The concentrations of glucose, xylose, and ethanol were determined using the Prominence LC-20A HPLC (Shimadzu, Kyoto, Japan) equipped with an Aminex HPX-87H ion-exchange column (Bio-Rad, Hercules, CA, USA) and a refractive index detector RID-10A. Samples were eluted from the column at 45 °C with 5 mmol L^−1^ H_2_SO_4_ at a flow rate of 0.6 mL min^−1^ [16].

### 2.6. Calculation of Physiological Parameters

The cell density (OD_600_) was determined with a UV–visible spectrophotometer (Eppendorf, Hamburg, Germany). The biomass was estimated according to the correlation of measured OD_600_ and dry weight [16]. One unit of OD_600_ equaled 0.208 g DCW L^−1^. The specific growth rate (μ) was the regression coefficient of the log-linear regression of the OD_600_ versus time during the exponential growth phase. The specific xylose consumption rate (r_xylose_), specific glucose consumption rate (r_glucose_), and specific ethanol production rate (r_ethanol_) were calculated using the following Equation (1) as previously described [12]:(1)$$r=\frac{A_{n}-A_{m}}{\frac{1}{2}\sum_{i=m+1}^{n}\left(B_{i}+B_{i-1}\right)\times \left(t_{i}-t_{i-1}\right)},$$
where r is the specific utilization or production rate during the phase from sampling point *m* to sampling point *n*; and *A*, *B*, and *t* are the metabolite concentration, biomass concentration, and time, respectively, at sampling points *n*, *i*, and *m*.

The *t*-tests were applied to evaluate the differences between means.

### 2.7. Quantitative PCR

qPCR data were analyzed according to the 2^−ΔΔ*C*t^ method [30]. RNA was extracted from exponentially growing cells using a UNlQ-10 Column Trizol Total RNA Isolation Kit (Sangon Biotech Co., Ltd., Shanghai, China). The cDNA was obtained using a PrimeScript^™^ RT reagent Kit (TaKaRa, Shiga, Japan). The gene transcription levels were determined using the equation N = 2*^C^*^t (reference gene)^/2*^C^*^t (target gene)^. *ACT1* was used as the reference gene, and t-tests were applied to evaluate the differences between means [16].

## 3. Results

### 3.1. Expression of the GPA2^G132V^ Allele and Deletion of Both PDE1 and PDE2 Increased PKA Activity in S. cerevisiae

To investigate differences in cAMP-PKA responses in *S. cerevisiae* to glucose and xylose, intracellular cAMP concentration and trehalase activity level of strains cultured in media using glucose or xylose as the sole carbon source were determined, since trehalase activity is tightly connected to PKA activity [31]. The results showed that intracellular cAMP concentrations of the strain BSL01 in glucose and xylose fermentation were similar (Figure 2). However, intracellular trehalase activity of yeast cells in xylose fermentation was only 35% of that in glucose fermentation (Figure 2), implying that the PKA activity of cells in xylose fermentation was lower than that in glucose fermentation.

Three genetic strategies were used to increase PKA activity. The first strategy promoted expression of the *GPA2^G132V^* allele, which can constitutively activate the cAMP-PKA pathway whether or not Gpr1 senses the extracellular glucose signals [21,32]. The second promoted expression of the *RAS2^G19V^* allele, which may induce high levels of cAMP and PKA activity [21,33]. In the third method, we deleted the genes encoding two phosphodiesterase (both Pde1 and Pde2 or only the primary one, Pde1), which functions in the feedback control by PKA in degrading cAMP [19]. Thus, we constructed strains BSL06(*GPA2^G132V^*), BSL08(*RAS2^G19V^*), BSL10(*pde1Δ*), and BSL16(*pde1Δ pde2Δ*), which were derived from the xylose-utilizing strain BSL01. The BSL01 strain consisted of a xylose isomerase gene cloned from a metagenomics library of the bovine rumen microbiome. Moreover, the genes encoding xylulokinase and the enzymes in the non-oxidative part of the PPP were overexpressed in BSL01.

The results show that expressing *GPA2^G132V^* and deleting both *PDE1* and *PDE2* increased the intracellular cAMP concentration by respectively 15% and 30% in glucose fermentation (Figure 2A). Consistent with this, trehalase activity increased by respectively 76% and 82% (Figure 2B), which implied the increase of PKA activity. In xylose fermentation, cAMP levels of the five strains were similar. However, expressing *GPA2^G132V^* and deleting both *PDE1* and *PDE2* increased trehalase activity by respectively 73% and 114%. Thus, the trehalase activity of these two strains in xylose fermentation respectively reached 61% and 70% of that of BSL01 in glucose fermentation (Figure 2B). In contrast, expressing *RAS2^G19V^* and only deleting *PDE1* had no effect on cAMP level and trehalase activity in both glucose and xylose fermentations (Figure 2).

### 3.2. Expression of GPA2^G132V^ Allele and Deletion of Both PDE1 and PDE2 Affected Glucose and Xylose Metabolism

The fermentation dynamics of strains BSL06(*GPA2^G132V^*), BSL16(*pde1Δpde2Δ*), and the control BSL01 were compared. The results revealed that the specific growth rates (μ) of these strains with 20 g L^−1^ xylose as the sole carbon source were the same (Figure 3). The specific xylose consumption rates (r_xylose_, xylose consumed per unit of cell mass) of BSL06(*GPA2^G132V^*) and BSL16(*pde1Δpde2Δ*) were 19% and 51% higher than that of BSL01, respectively. The specific ethanol production rates (r_ethanol_) of BSL06(*GPA2^G132V^*) and BSL16(*pde1Δpde2Δ*) were greater by 74% and 72%, respectively, compared to that of BSL01 (Figure 3, Table 2).

In the co-fermentation of 20 g L^−1^ glucose and 20 g L^−1^ xylose as carbon sources, the μ of strains were also similar (Figure 4). However, BSL06(*GPA2^G132V^*) and BSL16(*pde1Δpde2Δ*) shifted out of the exponential growth phase before BSL01′s shift. Correspondingly, biomass yields of BSL06(*GPA2^G132V^*) and BSL16(*pde1Δpde2Δ*) were lower than biomass of BSL01. Interestingly, the lower biomass yields did not result in lower consumption rates of sugars; in fact, sugar metabolism rates of the mutant strains were greater than the rate of the wild type. Compared to BSL01, the specific glucose consumption rate (r_glucose_) of BSL06(*GPA2^G132V^*) increased by 11.1%; r_xylose_ of BSL16(*pde1Δpde2Δ*) increased by 48%; and r_ethanol_ of BSL06(*GPA2^G132V^*) and BSL16(*pde1Δpde2Δ*) increased by 14% and 22%, respectively (Figure 4, Table 2).

### 3.3. Weak Responses of Rgt2 and Snf3 to High Concentrations of Extracellular Xylose

To investigate if the Rgt2/Snf3-Rgt1 pathway, another well-known glucose signaling pathway, can respond to extracellular xylose and induce expression of transporter genes, the transcriptional levels of genes *HXT1* and *HXT2* in yeast cells cultured on different carbon sources were measured by quantitative PCR.

A previous study reported that a high concentration of glucose (40 g L^−1^) induced expression of *HXT1*, and the main sensor of the high-level glucose in *S. cerevisiae* was Rgt2 [22]. In addition, a low concentration of glucose (1 g L^−1^) induced expression of *HXT2*, and the main sensor of the low-level glucose was Snf3 [22]. Our results supported these findings (Figure 5). *HXT1* and *HXT2* transcriptional levels in the wild type strain CEN.PK 113-5D grown in the medium containing 40 g L^−1^ glucose were approximately 400 and 25 times greater than those observed in cells cultured in a medium containing glycerol as the sole carbon source (hereafter referred to as glycerol medium), respectively. Furthermore, in comparison to *HXT1* expression in the wild type exposed to the high concentration of glucose, expression in *rgt2Δ* mutants was greater by 58% and unaffected by the deletion of *SNF3*. In a similar comparison, *HXT2* expression induced by 1 g L^−1^ glucose was lower by 82% in *snf3Δ* mutants and unaffected by the deletion of *RGT2.*

The results illustrate additional relationships of the four genes and the different carbon sources (Figure 5). The presence of 1 g L^−1^ glucose, 40 g L^−1^ xylose, or 1 g L^−1^ xylose also increased the transcriptional levels of *HXT1* in CEN.PK 113-5D, although the levels were much lower than the level in which 40 g L^−1^ glucose was present. Thus, all four carbohydrate conditions can induce the expression of *HXT1*. However, deletion of *RGT2* or *SNF3* reduced expression of *HXT1* in strains cultured in 1 g L^−1^ xylose, the weak response of the strains to the low xylose concentration (1 g L^−^^1^) may be a cumulative effect that involves these two sensors. Exposure to 40 g L^−1^ glucose and 40 g L^−1^ xylose also raised transcriptional levels of *HXT2* in CEN.PK 113-5D, although the levels were much lower than the exposure to 1 g L^−1^ glucose. By contrast, the signal from 1 g L^−1^ xylose was likely too weak to induce expression of *HXT2.* Furthermore, deletion of *SNF3* significantly weakened the expression of *HXT2* in the yeast cultured in 1 g L^−1^ glucose or 40 g L^−1^ xylose, while deletion of *RGT2* did not weaken expression of *HXT2*. These results suggest that *S. cerevisiae* mainly depends on Snf3 to sense a high concentration of xylose (40 g L^−1^), similar to its Snf3-dependent response to the low level of glucose, though the response was clearly much weaker than the response to glucose.

### 3.4. Deleting RGT1 Enhanced Xylose Utilization in S. cerevisiae by Upregulating the Expression of Hexose Transporter Genes

Rgt1, a transcription repressor of *HXTs*, is phosphorylated when the glucose signal is transmitted into the cell through Rgt2 and Snf3, and then released from the promoters of *HXTs*; therefore, it derepresses the expression of hexose transporters. Thus, to simulate the condition that glucose signals have occurred, *rgt1* was deleted in strain BSL01. The qPCR results indicate that deletion of *RGT1* increased transcriptional levels of *HXT1* and *HXT2* by 17.32 ± 2.23- and 3.75 ± 0.38-fold, respectively, in cells cultured with 20 g L^−1^ xylose, but *RGT1* had no notable effect on cells cultured with 20 g L^−1^ glucose. Furthermore, the results from fermentation experiments (Figure 6) showed that the specific xylose consumption rate (r_xylose_) of BSL20(*rgt1Δ*) in xylose fermentation was 0.133 ± 0.002 g L^−1^ h^−1^ g^−1^ DCW, which was 24% higher than the r_xylose_ of BSL01 (0.107 ± 0.006 g L^−1^ h^−1^ g^−1^ DCW). Consistent with qPCR results, the *RGT1* deletion did not affect glucose utilization. However, the r_xylose_ of BSL20(*rgt1Δ*) in the glucose–xylose co-fermentation during the xylose consumption phase (20–72 h) was 0.053 ± 0.003 g L^−1^ h^−1^ g^−1^ DCW and 23% higher than that of BSL01 (0.043 ± 0.003 g L^−1^ h^−1^ g^−1^ DCW).

## 4. Discussion

Xylose is an important and abundant carbon source second to glucose in industrial fermentation processes to produce fuels and chemicals. However, *S. cerevisiae* is only able to efficiently metabolize glucose. Despite the many attempts to genetically modify the carbohydrate metabolism of yeast to obtain more efficient xylose fermentation rates, large gaps in our knowledge and understanding of the carbohydrate metabolic pathways and their mechanisms hamper our pursuit in developing recombinant strains that can exploit the potential energy from xylose as well as they do from glucose.

Hexose transporters and glycolysis, which are controlled by glucose signaling pathways, are important to xylose metabolic efficiency, and our previous work suggests that extracellular glucose signals can promote xylose utilization [18] (Figure A1). Despite extensive studies on glucose signaling pathways and their controls on glucose metabolism, less is known about how these signaling pathways may respond to xylose. Here, we demonstrated that *S. cerevisiae* cells can only weakly respond to xylose, which may limit its potential to metabolize xylose. Furthermore, we also demonstrated that simulating the occurrence of glucose signals in yeast strains with a mutation (*GPA2^G132V^*) or deletions (*PDE1* and *PDE2*, and *RGT1*) of specific genes found in two different glucose signaling pathways (cAMP-PKA and Rgt2/Snf3-Rgt1 pathways) are feasible strategies to enhance xylose utilization and/or ethanol production in yeast.

Brink et al. [25] developed a successful method by applying a green fluorescence protein gene (yEGFP3) to monitor effects of eight endogenous yeast promoters, which are found in the three major sugar metabolism pathways Snf3/Rgt2, Snf1/Mig1, and cAMP/PKA, on glucose and xylose signaling responses in yeast. They determined that only two of these promoters respond to xylose; the two successful promoters were the *HXT2p/4p* sensors of the Snf3/Rgt2 signaling pathway. We also measured the responses of the two promoters, *HXT1* and *HXT2*, but in *rgt2Δ* and *snf3Δ* mutants, as well as a strain with the wildtype Snf3/Rgt2 pathway. Our results revealed that the expression of *HXT1* responded to both high and low concentrations of xylose (50 and 1 g L^−1^, respectively), while the expression of *HXT2* only responded to the high xylose concentration. Our results confirmed that the Snf3/Rgt2 pathway can minimally respond to xylose. Furthermore, we found that *S. cerevisiae*’s response to the high xylose level is similar to its response to the low glucose level, which corroborates the findings of Osiro et al. [34].

Özcan et al. (1996) revealed that a dominant mutation in *RGT2* and *SNF3* causes constitutive induction of HXT gene expression, even in the absence of the inducer glucose, which suggests this mutation converts the glucose sensors into their glucose-bound form [22]. Furthermore, their study on the *rgt2Δsnf3Δ* strain, as well as the *rgt2Δ* and *snf3Δ* strains, revealed that *HXT1* and *HXT2* took even lower response to the glucose in the double mutant than in the single mutants, which confirmed that Snf3 and Rgt2 have separate but overlapping functions [35]. In the present work, deletion of either *RGT2* or *SNF3* reduced expression of *HXT1* in strains cultured in 1 g L^−1^ xylose, which suggest that the weak response of the strains to the low xylose concentration (1 g L^−1^) may be a cumulative effect that involves these two sensors. Moreover, xylose can bind both Snf3 and Rgt2 and slightly alter their conformations. In contrast, deletion of *SNF3* significantly weakened the expression of *HXT2* in the yeast cultured in 40 g L^−1^ xylose, while deletion of *RGT2* did not weaken expression of *HXT2*, suggesting that *S. cerevisiae* mainly depends on Snf3 to sense a high concentration of xylose (40 g L^−1^). Our results also showed that the response of cells to xylose was clearly much weaker than the response to glucose. This may be due to either a low affinity of xylose to Snf3/Rgt2 is low or the conformations of xylose-bound Snf3/Rgt2 are not close to the glucose-bound Snf3/Rgt2. To determine the cause, will require substantially greater research efforts in the future.

Here, we used a recombinant strain with xylose metabolic capacity. This strain was derived from CEN.PK-113-5D, which belongs to the family of the CEN.PK strain [27]. Reportedly, the *CYR1* allele in CEN.PK strains encodes a mutant adenylate cyclase (Cyr1^K1876M^), and the constitutive activation of the cAMP pathway by Ras2^G19V^ and Gpa2^G132V^ is suppressed by this mutant [27]. Supporting this report, our result showed that the cAMP level and PKA activity of strain BSL08(*RAS2^G19V^*) were not higher than those of the control [36]. However, although the cAMP level of the strain BSL06(*GPA2^G132V^*) showed an insignificant increase in xylose, the increased PKA activity in BSL06(*GPA2^G132V^*) cells cultured on glucose and xylose were observed. This may be due to Gpa2 inhibiting Gpb1 and 2, which inhibit PKA activity [37]. Moreover, deletion of both *PDE1* and *PDE2* increased PKA activity and xylose utilization. Recent works have reported that deletion of *IRA2* inhibits PKA activity by negatively regulating RAS or *BCY1* (a gene encoding a PKA inhibitor) and thus increases the specific xylose consumption rates (xylose consumed per unit of cell mass) of strains [14,15,38]. Our results directly support their viewpoint that active PKA benefits xylose metabolism in a certain way.

Moreover, we found that increasing PKA activity did not affect the specific growth rate of the yeast strain; however, it shortened the exponential growth phase and decreased biomass yield in glucose fermentation. High PKA activity promotes aging and early cell death and leads to a short lifespan [39,40], and thus the high PKA activity may have been the cause of this observation. On the other hand, in xylose fermentation, the increased PKA activity did not affect the exponential growth phase and biomass yield of yeast. This may be due to that the PKA activity in strains (control and mutations) cultured in xylose are lower than that in the control strain cultured in glucose. However, the short lifespan should be avoided in ethanol production using materials containing both glucose and xylose. Therefore, increase PKA activity only when glucose is running out in glucose–xylose co-fermentation may be a way to enhance xylose utilization without shorten the lifespan of yeast cells.

## Figures and Tables

**Figure 1 microorganisms-08-00100-f001:**
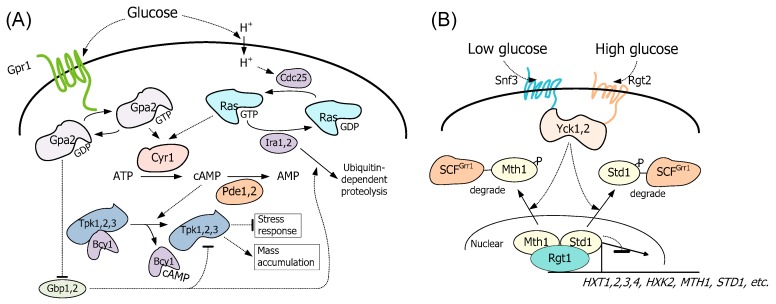
Two signaling pathways that respond to extracellular glucose in *Saccharomyces cerevisiae*. (**A**) The cAMP-PKA pathway. The membrane protein Gpr1 serves as a receptor for extracellular glucose and stimulates activation of Gpa2, which in turn stimulates adenylyl cyclase Cyr1. Ras responds to glucose-stimulated intracellular acidification and also stimulates Cyr1. Cellular levels of cAMP were determined by the competing activities of synthesis from ATP via Cyr1 and degradation to AMP by phosphodiesterases, Pde1, and Pde2. A high level of cAMP activates PKA by binding to the regulatory subunits Bcy1 which releases the catalytic subunits Tpk1, 2, 3. Additionally, Gpa2 can activate PKA by inhibiting Gpb1, 2, which inhibits PKA and promotes ubiquitin-dependent proteolysis of Ira2. The active PKA suppresses the stress response and stimulates growth. (**B**) The Snf3/Rgt2-Rgt1 pathway. High and low levels of extracellular glucose are sensed by the membrane-spanning proteins Rgt2 and Snf3, respectively. Glucose binds to the glucose receptors Snf3 and Rgt2 and stimulates the Yck kinases that phosphorylate Std1 and Mth1. The phosphorylated Std1 and Mth1 are then degraded by ubiquitin-dependent proteolysis. Without corepressors Std1 and Mth1, Rgt1 does not repress the expression of genes such as *HXTs*, etc.

**Figure 2 microorganisms-08-00100-f002:**
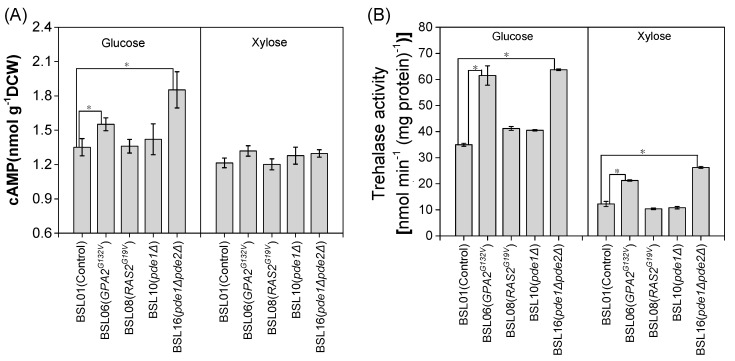
Amounts of cAMP (**A**) and trehalase activity (**B**) of recombinant *S. cerevisiae* strains. Cells with an initial OD_600_ of 1.0 were cultured at 30 °C in shake flasks at 200 rpm. Cells were sampled at mid-exponential growth phase, specifically at 12, 12.5, and 13 h of xylose fermentation; and at 4.5, 5, and 5.5 h of glucose fermentation. At each time point, three samples were taken, one from each of the triplicate fermentations. Data are mean values ± standard deviations of the three samples collected per time point. * *p* value < 0.05.

**Figure 3 microorganisms-08-00100-f003:**
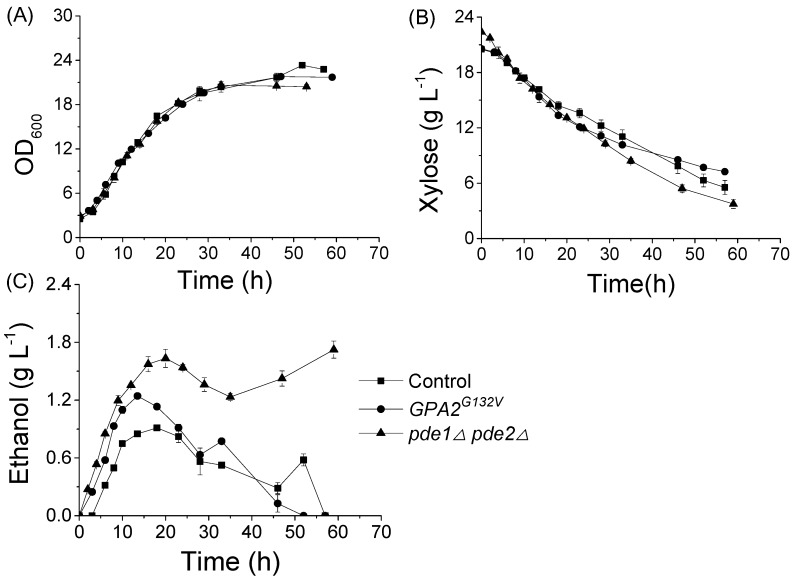
Xylose fermentation characteristics of recombinant *S. cerevisiae* strains. The (**A**), growth; (**B**) xylose consumption; and (**C**), ethanol production of strains. BSL06(*GPA2^G132V^*) (●), BSL16(*pde1Δ pde2Δ*) (▲), and the control BSL01 (■). Cells with initial OD_600_ of 2.5 (≈0.5 g L^−1^ biomass) were cultured in SC-URA medium supplemented with 20 g L^−^^1^ xylose at 30 °C and 200 rpm in shake flasks. Data are the mean values of triplicate tests.

**Figure 4 microorganisms-08-00100-f004:**
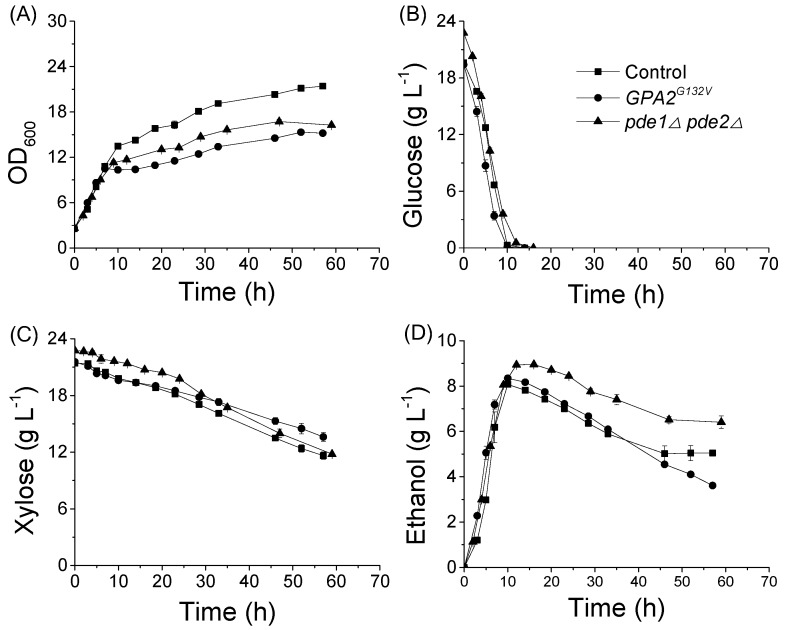
Xylose and glucose co-fermentation characteristics of recombinant *S. cerevisiae* strains. The (**A**), growth; (**B**) glucose consumption; (**C**), xylose consumption; and (**D**), ethanol production of strains. BSL06(*GPA2^G132V^*) (●), BSL16(*pde1Δ pde2Δ*) (▲), and the control BSL01 (■). Cells with initial OD_600_ of 2.5 (≈0.5 g L^−1^ biomass) were cultured in SC-URA medium supplemented with 20 g L^−1^ xylose and 20 g L^−1^ glucose at 30 °C and 200 rpm in shake flasks. Data are the mean values of triplicates.

**Figure 5 microorganisms-08-00100-f005:**
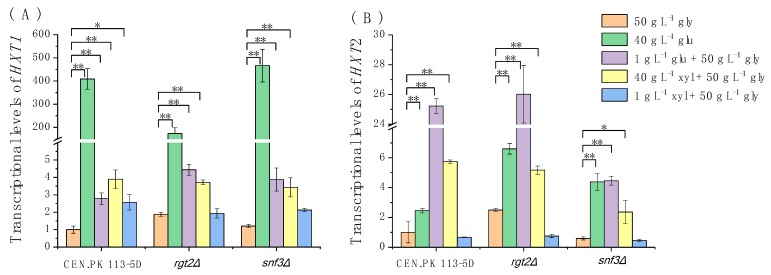
The transcriptional levels of (**A**) *HXT1* and (**B**) *HXT2* in yeast strains. The gene transcript levels in CEN.PK 113-5D (wild type) cells cultured in 50 g L^−1^ glycerol were defined as 1 and all other data were relativized to the level of the wild type cultured in 50 g L^−1^ glycerol. Abbreviations: gly, glycerol; glu, glucose; xyl, xylose. Cells with initial OD_600_ of 0.2 were cultured at 30 °C in shake flasks that were agitated at 200 rpm. Cells were collected when the OD_600_ reached 0.8–1.0 and then their mRNA was extracted. The experiments were performed in triplicate. *, fold change ≥ 2, *p* < 0.05; **, fold change ≥ 2, *p* < 0.005.

**Figure 6 microorganisms-08-00100-f006:**
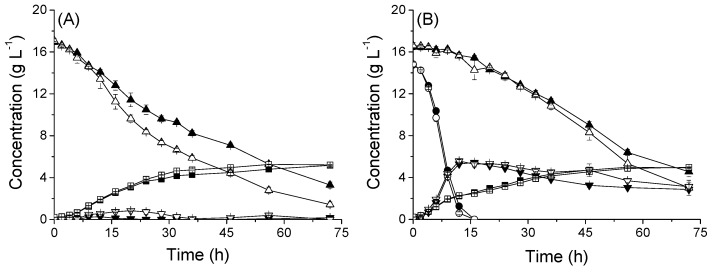
Fermentation characteristics of *RGT1* deletion strain. Strains cultured in (**A**) 40 mL SC-URA with 20 g L^−1^ xylose or (**B**) 40 mL SC-URA with 20 g L^−1^ xylose and 20 g L^−1^ glucose. Cells with initial OD_600_ of 1 (≈0.2 g L^−1^ biomass) were agitated at 200 rpm at 30 °C. The experiments were performed in triplicate. Symbols: ■□, dry cell weight; ●○, glucose; ▲△, xylose; ▼▽, ethanol. Solid symbols represent the BSL01 strain and hollow symbols represent the mutant strain BSL20 (*rgt1Δ*).

**Table 1 microorganisms-08-00100-t001:** Strains and plasmids used in this work.

*S. cerevisiae* Strains and Plasmids	Description	Sources
**Plasmids**		
pUG6	*E. coli* plasmid with segment *LoxP-KanMX44-LoxP*	[28]
pJFE1	YCplac33; *CEN*, *Amp^r^*, *URA3*, *TEF1p-PGK1t*	[26]
pJX7	ΥΕplac195; 2μ, *Amp^r^*, *URA3*, *TEF1p-Ru-xylA-PGK1t*	[26]
YEp-CH	YEp; containing hygromycin B resistant gene and *Cre* gene under *GAL2* regulative regulation promoter	[26]
***S. cerevisiae* Strains**		
CEN.PK 113-5D	*MATa*; *ura3-53*, belongs to CEN.PK strain family	[27]
BSPC039	CEN.PK113-5D derivative; *XKS1(-194,-1)::loxP-P_TEF1_, gre3(-241, +338)::P_TPI1_-RKI1-T_RKI1_-P_PGK1_-TAL1-T_TAL1_-P_FBA1_-TKL1-T_TKL1_-P_ADH1_-RPE1-T_RPE1_-loxP*	[12]
BSL01	BSPC039 derivative; pJX7	This work
BSL06	BSL01 derivative; *GPA2^G132V^*	This work
BSL08	BSL01 derivative; *RAS2^G19V^*	This work
BSL10	BSL01 derivative; *pde1Δ*	This work
BSL16	BSL01 derivative; *pde1Δ*; *pde2**Δ*	This work
BSL20	BSL01 derivative; *rgt1Δ*	This work
BSWW1	Derivative of CEN.PK 113-5D: *snf3Δ*	This work
BSWW2	Derivative of CEN.PK 113-5D: *rgt2Δ*	This work

**Table 2 microorganisms-08-00100-t002:** The xylose fermentation and glucose–xylose co-fermentation characteristics of strains.

Strains	Xylose Fermentation		Glucose–Xylose Co-Fermentation		
	Specific Xylose Consumption Rates (g L^−1^ h^−1^ g^−1^ DCW)	Specific Ethanol Production Rates (g L^−1^ h^−1^ g^−1^ DCW)	Specific Glucose Consumption Rates (g L^−1^ h^−1^ g^−1^ DCW)	Specific Xylose Consumption Rates (g L^−1^ h^−1^ g^−1^ DCW)	Specific Ethanol Production Rates (g L^−1^ h^−1^ g^−1^ DCW)
BSL01(Control)	0.103 ± 0.003	0.047 ± 0.002	1.218 ± 0.004	0.046 ± 0.002	0.428 ± 0.005
BSL06(*GPA2^G132V^*)	0.123 ± 0.003	0.082 ± 0.001 ***	1.354 ± 0.019 **	0.0481 ± 0.002	0.489 ± 0.002 ***
BSL16(*pde1Δ pde2Δ*)	0.156 ± 0.001 **	0.081 ± 0.003 **	1.260 ± 0.008 *	0.068 ± 0.004 *	0.523 ± 0.005 ***

The fermentations were performed in the 100 mL shake flasks with 40 mL medium. For the xylose fermentation, the medium was SC-URA with 20 g L^−1^ xylose as carbon source; for the glucose–xylose co-fermentation, the medium was SC-URA with 20 g L^−1^ glucose and 20 g L^−1^ xylose medium. The cells were cultured at 30 °C, 200 rpm, with initial OD_600_ of 2.5 (≈0.5 g L^−1^ biomass). The experiments were performed in triplicate. The values are given as the averages ± standard deviations of three measurements. DCW, dry cell weight. * *p* value < 0.05; ** *p* value < 0.01; *** *p* value < 0.005.

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
