# Peer review of "Simulating Extracellular Glucose Signals Enhances Xylose Metabolism in Recombinant Saccharomyces cerevisiae"

_microorganisms, 2020, doi:10.3390/microorganisms8010100_

Round 1

Reviewer 1 Report

Manuscript microorganisms-657312, “Simulating extracellular glucose signals enhances xylose metabolism in recombinant Saccharomyces cerevisiae” by Wu et al.

Manuscript summary Xylose is a major sugar in plant biomass. There Is significant interest in converting xylose into renewable biofuels. However, the yeast Saccharomyces cerevisiae, the predominant industrial biocatalyst, is unable to sufficiently metabolize xylose without directed engineering and evolution. Previously, the authors determined that activation of glucose signaling pathways may enhance xylose metabolism. The authors followed up on this observation by examining the effects of activating glucose signaling pathways on xylose metabolism, and investigating the influence of xylose on glucose signaling pathways. The authors will need to address major issues before the manuscript is acceptable for publication. One major concern for this paper is that activation of the PKA pathway has been shown to enhance xylose metabolism by others (Sato et al., 2016, doi: 10.1371/journal.pgen.1006372; Myers et al., 2019, doi: 10.1371/journal.pgen.1008037; Osiro et al., 2019, doi: 10.1371/journal.pgen.1008037; Wagner et al., 2019, doi: 10.1371/journal.pone.0212389), so the authors will need to provide additional results beyond what is currently in the manuscript. Additionally, a major conclusion of the manuscript is that the gpa2G132V mutation is a feasible strategy to enhance xylose utilization, however, the effect of the mutation is not statistically significant (see Major Comment #3). Lastly, the Discussion lacks clear interpretation of results, explanations for observations that are inconsistent with published results and substantive insight into the connection between xylose metabolism and glucose signaling pathways. I recommend that the authors significantly revise and resubmit their manuscript.

Major comments and concerns 1. To determine whether xylose sensing also utilizes the PKA signaling pathway, the authors examine both intracellular cAMP concentrations and trehalase activities in various mutant strains exposed to xylose. While mutants with activated PKA pathways display expected changes in trehalase activities, there are no significant differences in cAMP levels (Fig. 2A) in xylose medium and the authors do not provide a direct interpretation of this result. Can the authors provide an explanation for this discrepancy? Why did the authors observe no effects with the rasG132V mutant in glucose? 2. The reproducibility of ethanol production in Fig 3 is surprising for an experiment done in independent triplicates. Can the authors explain the decrease in ethanol titer between 20-35 hrs by the pde1 D pde2 D double mutant, the jump in ethanol titer at 35 hr for the gpa2G132V mutant, and the jump in ethanol titer at 54 hr for the control strain? It is more surprising that these fluctuations would reproduce in independent experiments, which this reviewer would define as experiments performed on separate days. This reviewer suspects that these fluctuations in ethanol titer are likely due to analytical errors. If the authors performed these triplicate experiments at the same time, they should remove “independent” from “independent triplicate tests” in the figure legend. 3. In lines 219-221, the authors state that the specific xylose consumption rate for BSL06 is 19% faster than BSL01. However, Table 2 does not indicate that this result is statistically
significant. Table 2 also indicates that the specific xylose consumption rate for BSL06 is not statistically different from BSL01 in 20 g/L glucose + 20 g/L xylose medium. Therefore, it is unclear that Gpa2 activation impacts xylose metabolism, invalidating the authors conclusion that “…BSL06 showed increased PKA activity…on xylose” (Lines 377378). The authors should perform additional fermentation experiments (at least two more independent biological replicates, not performed on the same day) to gain statistical power. 4. The model in Fig 1 indicates that both RGT2 and SNF3 regulate HXT1 and 2 expression in response to glucose. The data presented in Fig 5. does not indicate this; RGT2 is important for HXT1 but not HXT2 expression in 40 g/L glucose, while deletion of SNF3 has no effect on HXT1 expression in any condition. Additionally, expression of HXT2 in the rgt2 D mutant was increased in 40 g/L glucose and had no effect in 1 g/L glucose. The authors need to explain these inconsistencies in their results with what has been published. Authors should also provide statistical significance for comparisons across genotypes in Fig 5 (in addition to comparing across media conditions). As such, the conclusion that the authors “results confirmed that the Snf3/Rgt2 pathway can minimally respond to xylose” (Line 364-365) is not justified. 5. Lines 253-255, the authors suggest that the weak response of the strains to the low xylose concentration is a cumulative effect of the two sensors. The authors should perform the simple experiment to test this by constructing a rgt2 D snf3 D double mutant, growing the strain in xylose medium and measuring HXT1 and HXT2 mRNA levels. 6. In the Discussion (Lines 384-392), the authors try to connect PKA activity with shortened exponential growth phase and decreased biomass yield to shortened lifespan (Longo and Fabrizio, 2012, doi: 10.1007/978-94-007-2561-4_5). This connection is unclear; the paper by Longo and Fabrizio states that lifespan is “a measure of the mean and maximum survival time of non-dividing yeast populations while the replicative life span is based on the mean and maximum number of daughter cells generated by an individual mother cell before cell division stops irreversibly.” Is the shortened exponential growth phase observed by the authors due to loss of survival, inability to replicate, or simply due to the depletion of nutrients? In order to reasonably make this suggestion, the authors should provide data or additional supporting references.

Additional comments 1. Line 46, “…build metabolic pathways of xylose.” This should be corrected to “…build pathways for xylose metabolism.” 2. Line 54. The authors reference five papers (two of which are their own) describing evolved yeast strains that can utilize xylose. There are many more papers that describe the generation of yeast strains evolved for xylose metabolism. 3. Line 58 an7d 6, “…signal pathway…” changed to “…signaling pathway…” 4. Line 61, “…into its cells,” to “…intracellularly,”. 5. Lines 67-76 and lines 79-81 lacks any references. The authors need to include references. 6. Lines 88-89 lacks any references. The authors need to include references.
7. Line 98, “…increased the PKA level…” As written, this says that PDE1 and PDE2 increased the PKA protein level. Do the authors mean, “…increased PKA-regulated expression…”? 8. Lines 192-193, “The results showed that…were similar.” The authors need to refer to Fig 2. 9. Line 194, “implied” should be changed to “implying”. 10. Line 215, GPA2G132V, PDE1 and PDE2 should be changed to gpa2G132V, PDE1 and PDE2. Mutants should be lowercase and in italics, wildtype genes should be capitalized. 11. Lines 238-239, “A previous study…was Rgt2.” This needs to have a reference. 12. Line 299. Were the “triplicate fermentations” done in independently, i.e., on three separate days? 13. Lines 300, 318, 335-336, authors should describe how the p values were calculated. 14. Line 333, “with20” should be “with 20”. 15. Line 355. Ethanol production is not truly addressed by the authors, since all fermentations were done aerobically. 16. Line 378, “This may due to Gpa2…” should be corrected to, “This may be due to Gpa2 inhibiting Gpb1 and 2, which inhibit PKA activity.” 17. Line 379, “Moreover, deletion both PDE1 and PDE2 did increase…” should be corrected to, “Moreover, deletion of both PDE1 and PDE2 increased…”?

Author Response

Thank you for your detailed suggestions, our responses are list below.

To major comments and concerns:

1. In fact, we've explained the problem, but it's clearly not clear enough. In the revised version, we have explained it in more detail and hope it can be accepted. (Line 373-378).

About no effects was observe with the rasG132V mutant in glucose, we proposed that the PKA have been activated in cells cultured in glucose, this mutation did not give an effect beyond the activated level, therefore, no effect could be observed. However, we do not want to show this propose in the manuscript without supporting data.

2. The fermentation experiments were performed in the same day, three separate colonies cultured in three separate shake flasks. The “independent” was removed.

3.We did the independent biological replicates, not in a day. The increase also observed. However, the p value just a little bit higher than 0.05.

4. We are sorry for the uncleared description of the Snf3/Rgt2-Rgt1 pathway. We modified the Fig 1 and add explanation in the legend, which sentence also appears in the introduction section (Line 81-83). Since Snf3 and Rgt2 response to different level of sugars, therefore the comparison of across genotypes is not necessary and will not affect our conclusion.

5. The rgt2Δ snf3Δ double mutant could be another piece of evidence.

6. This is a good suggestion. The shortened exponential growth phase and decreased biomass yield we observed may be an outcome of combined effects. As far as we know how the PKA affect two kinds of lifespans are still questions for people study on the lifespan. Many works need to do for details.

To additional comments

2. There are many more papers should be reference as reviewer suggested. That is hard to choose. That’s why we apologize to colleagues whose work could not be cited in the Acknowledgments.

10. We write those as that in the references.

15. Saccharomyces cerevisiae is a kind of Crabtree-positive yeast. It produces ethanol if the concentration of glucose is high independent the oxygen conditions. When xylose is the carbon source, the accumulate of ethanol depends on the consumption rate of xylose. Therefore, the ethanol production in aerobically fermentations is meaningful.

Others are modified as reviewer suggested, thank you.

Reviewer 2 Report

This manuscript is a revised version of the manuscript submitted some months ago to another journal and for which I acted as reviewer. Although some comments have been taken into consideration (mostly removing data that do not match with the conclusions and adding a few other references than their own), the current version remains very similar to the previous one. A major conclusion of the revised discussion is that PKA is an engineering target. But this is only confirmatory since the link between PKA and xylose sensing was already highlighted by other recent papers: Wagner et al (/doi.org/10.1101/540534), Osiro et al (doi.org/10.1186/s12934-019-1141-x). Moreover PKA overexpression is not the solution since it leads to many negative side-effects, such as low biomass, low stress tolerance, etc. Another major conclusion of the abstract is that xylose consumption rate is increased by 24% in the rgt1 mutant; however in figure 6 (old figure 7), we see instead lower xylose consumption for rgt1 and the only difference is that there is equal amount of biomass formed from less xylose. One difference with previous studies is that they seem to observe sensing of extracellular xylose via the Rgt2/Snf3 route. This part is novel and should constitute the core of the study, the other parts being confirmatory.

Author Response

Thank you for pointing out the highlights and shortcomings of our work. We have tried to improve the manuscript.

About Figure 6, we are so sorry that the rgt1Δ and control were mislabeled each other, and since we believed that our result was descripted correctly, we did not realize this mistake even you show you doubts. They are corrected in the revised version. Thank you very much.

Reviewer 3 Report

This paper shows the importance of the gene PDE1 and PDE2, which encodes phosphoprotein phosphatase, in xylose fermentation of yeast. The data that authors calculated with the mathematical formula in the fermantation rate of Xylose are also interesting with the novel approch. I do not know this apprch in xylose fermentation. This result may be contributed the novel fermentation process of xylose. I have a little unkown letters which I want to edit in this paper, but I think this paper is suitable to publish.

Author Response

Thank you!

Round 2

Reviewer 1 Report

The authors did not adequately address major comments #5 and 6 by this reviewer.

Data from technical replicates of fermentation experiments are not the proper way to adequately show biological reproducibility and statistical significance. I will assume that the authors performed independent experiments that support the results shown in Figs. 2-6.

The manuscript contains some minor grammatical and spelling errors that can be corrected with editorial help.

Reviewer 2 Report

The authors did not change the angle of their study. PKA as a target is not
new. Instead they should focus around the signaling they observe in the
other route. This requires a major rewriting in the discussion section,
which is not there yet.
